# Assessment of the Bioactive Compounds in White and Red Wines Enriched with a *Primula veris* L.

**DOI:** 10.3390/molecules24224074

**Published:** 2019-11-11

**Authors:** Maria Tarapatskyy, Ireneusz Kapusta, Aleksandra Gumienna, Czesław Puchalski

**Affiliations:** 1Department of Bioenergetics and Food Analysis, Faculty of Biology and Agriculture, University of Rzeszów, 35-601 Rzeszów, Poland; gumienna.ola@wp.pl (A.G.); cpuchal@ur.edu.pl (C.P.); 2Department of Food Technology and Human Nutrition, Faculty of Biology and Agriculture, University of Rzeszów, 35-601 Rzeszów, Poland; ikapusta@ur.edu.pl

**Keywords:** wine, *Primula veris* L., cowslip, health potential, flavonoids, anthocyanins, triterpenoid saponins, antioxidant activity

## Abstract

The aim of this paper was to analyze selected physicochemical properties and the pro-health potential of wines produced in southeastern Poland, in the Subcarpathian region, and commercial Carlo Rossi wines enhanced with cowslip (*Primula veris* L.). This study used ultra-performance reverse-phase liquid chromatography (UPLC)-PDA-MS/MS to perform most of the analysis, including the polyphenolic compounds and saponin content in wines enriched by *Primula veris* L. The initial anthocyanin content in Subcarpathian (Regional) red wine samples increased four times to the level of 1956.85 mg/L after a 10% addition of *Primula veris* L. flowers. For white wines, a five-fold increase in flavonol content was found in Subcarpathian (Regional) and wine samples, and an almost 25-fold increase in flavonol content was found in Carlo Rossi (Commercial) wine samples at the lowest (2.5%) *Primula veris* L. flower addition. Qualitative analysis of the regional white wines with a 10% *Primula veris* L. flower enhancement demonstrated the highest kaempferol content (197.75 mg/L) and a high quercetin content (31.35 mg/L). Thanks to wine enrichment in triterpenoid saponins and in polyphenolic compounds from *Primula veris* L. flowers, which are effectively extracted to wine under mild conditions, both white and red wines can constitute a highly pro-health component of diets, which is valuable in preventing chronic heart failure.

## 1. Introduction

Wine is an alcoholic beverage produced as a result of alcoholic fermentation of fresh *Vitis vinifera* grapevine fruits (fragmented or not) or must. There are multiple wine classifications, e.g., by color, strength, or total sugar content. The decisive factor affecting the flavor, aroma, and composition of wine is the variety of the grapevine from which the wine is made. Other factors are natural environment characteristics, such as soil, climate, method of cultivation, vinification process technology, and many others [1]. 

The main components of wine are water (approximately 86%) and ethyl alcohol (12% on average), accompanied by much smaller amounts of glycerol, sugars, organic acids, mineral compounds, vitamins, tannins, and biologically active compounds, which are also responsible for wine’s pro-health properties, such as its anti-oxidant, anti-carcinogenic, anti-inflammatory, immunomodulating, anti-virus and anti-bacterial effects [2,3]. It has also been demonstrated that moderate wine intake greatly reduces the risk of cardiovascular diseases, although due to the alcohol content, international guidelines limit wine consumption to the level of 150–250 mL (15–30 g alcohol) [2].

The increased interest in wine production in Poland is related to climate changes (a rising trend in temperature and shorter winters have been observed), the emergence of more resistant grapevine varieties, greater knowledge concerning grapevine cultivation, increased consumer awareness of wine’s pro-health properties, and farmers seeking new, profitable crops [4,5]. 

Grapevine cultivation in Poland is not the easiest task as the climate differs significantly from the conditions that are characteristic of typical viticultural regions. Despite this, winemaking is a growing industry in Poland, where the tradition reaches as far as the Middle Ages [4]. The climate in Poland is characterized by significant daily and seasonal fluctuations in temperature, with a risk of frost and hailstorms during the spring season [5]. The European Union has classified Poland as viticultural Zone A, i.e., the coldest one, together with Germany (except Baden), the Czech Republic (excluding Moravia), Belgium, and the United Kingdom [6], which means that wines from Poland can be sold on the European market. 

The Subcarpathian region, located in the southeastern part of the country (number 6 in Figure 1) is considered exceptionally important for winemaking in Poland. At present, close to 150 vineyards are located there, and most of them environmentally friendly, whose total area is estimated at over 100 hectares [7,8]. Thanks to specific terrain features (foothills with gentle slopes, clay soils, poorly industrialized) and microclimate (hot summers and sunny autumns), the region has become a perfect location for the development of winemaking, as well as winemaking tourism [8]. 

In Poland, grapevine varieties that are hybrids of *Vitis vinifera* and species originating from North America (*Vitis labrusca* and *V. aestivalis*, so-called American hybrids; or *V. rupestris* and *V. lincekumii*, so-called French-American hybrids), or Asia (e.g., *V. amurensis*, the Amur grape) are the most commonly cultivated. These varieties are characterized not only by resistance to low temperatures but also to diseases, as well as high and regular yields [9]. The wines they produce exhibit a specific flavor and aroma, which stem from the slightly different chemical composition than wines produced from *V. vinifera.* The colder climate and shorter summers lead to grape fruits usually having a lower sugar content and higher acidity. Cold-climate wines are viewed as subtler and more “refined”, and the higher acidity imbues them with freshness [9]. For this reason, winemaking in the Subcarpathian region continues to grow, and the local wines are unique products that enjoy growing recognition on the market. As winemaking technology in Poland has evolved across decades, alternative uses for wine have been sought, particularly due to its pro-health properties. One such procedure has been enhancing wine with cowslip (*Primula veris* L.), which was supposed to enrich the wine with substances of high biological activity.

Polyphenolic compounds are commonly known as plant secondary metabolites that hold an aromatic ring bearing at least one hydroxyl group. These phytochemical substances are presented in nutrients and herbal medicines; many studies have reported on both flavonoids and other phenolic components due to their effective antioxidant, anticancer, anti-inflammation, and antibacterial properties; their function as cardioprotective agents; their immune system-promoting and skin protection from ultraviolet (UV) radiation factors, and their interesting possibilities for pharmaceutical and medical applications [10,11,12].

Over the last years, polyphenol-rich foods and polyphenols have received great attention due to their potential beneficial effects toward human health. Contained not only in fruits and vegetables, but also in whole grains, nuts, olive oil, and beverages such as coffee and tea, they are characteristic components of healthy dietary patterns. Recent evidence has proposed that a higher dietary intake of polyphenols may be inversely associated with overall and cardiovascular disease-related mortality, certain cancers, cardiovascular diseases, anthropometric measures, and mood disorders [13]. These features make polyphenols a potentially interesting material for the development of functional foods or possible therapy for the prevention of some diseases. The health effects of polyphenols depend on both their respective intakes and their bioavailability, which can vary greatly. Numerous genetic, environmental, and technologic factors may affect the polyphenol concentrations in food, some of which can be controlled to optimize the polyphenol content of foods. One of the possibilities for the increase of phenolic compounds in food may be the fruits and vegetables cultivar selection, as cultivars differ greatly in their phenolics content. The strong potential may also be the processing technology polyphenol fortification of food products, which must be taken into consideration.

Cowslip (*Primula veris* L.) is a plant species of the Primulaceae family. Until 2014, this plant was protected in Poland. According to the literature data, it is an herbal material that is rich in saponins (approximately 60%), including primulic acid 1 (PA 1), as well as numerous flavonoid compounds and flavonols, i.e., rutin, catechin, kaempferol, and luteolin [14,15]. In folk medicine, cowslip decoctions had been used as a sedative in migraine, insomnia, nervous stress, and menstruation ailment treatment. The diuretic properties of this herb had also been known, which led to its use in organism purging and detoxification treatments. Cowslip-enhanced wine, in turn, has been recommended for regulating blood circulation and in post-stroke and post-hemorrhage recovery. It is suspected that these properties may also be the result of the coumarin derivatives present in the herb. Cowslip’s bioactive substances have found use in pharmaceutical preparations, e.g., in mucoactive syrups and in preparations used in respiratory system diseases [16,17]. As Poland does not belong to leading wine producers, the production of specialist wines enhanced with pro-health additions in the form of green plants rich in bioactive agents is worthy of consideration. Therefore, the aim of this paper was to analyze selected physicochemical properties and the pro-health potential of wines produced in southeastern Poland, in the Subcarpathian region, and commercial Carlo Rossi wines enhanced with cowslip (*Primula veris* L.). Furthermore, polyphenolic compounds and saponins in herb-enhanced wine samples were identified. The tests were intended to determine the effectiveness of pro-health substance extraction in different wines and at different cowslip (*Primula veris* L.) content levels. Additionally, ethanol blends of different strengths were analyzed to study the effects of alcohol content on pro-health substance extraction from cowslip. An analysis of pro-health substance content may aid in determining the cardioprotective effects of white and red wines enhanced with *Primula veris* L., whose consumption may constitute an effective practice in preventing chronic heart failure. 

## 2. Results

Based on MS chromatography analysis, profiles of compounds belonging to anthocyanins (28), flavonols (22), and saponins (3) have been identified in the red wine samples, as well as 22 flavonols and three saponins in white wine samples. All the compounds identified are shown in Table 1 and Table 2, while Table 3 presents the concentrations of individual compounds, the polyphenolic compounds of the anthocyanin group, as well as their total content in regional and commercial red wines enhanced with *Primula veris* L. 

Anthocyanins constituted the dominant class of polyphenolic compounds in the test red wines, and their content ranged from 492.25 mg/L in Polish regional wines to 682.00 mg/L in popular Californian wines. Despite the higher total anthocyanin content in the test commercial wines, their profile differed from the regional wines in the lower content of delphinidin, cyanidin, and petunidin derivatives. Furthermore, a major drop in anthocyanin content was observed in all *Primula veris* L. enhanced commercial wine samples after 14 days of extraction, and the color of commercial wine blends enhanced with *Primula veris* L. flowers became brighter. The greatest drop in anthocyanin content, approximately six-fold, was found in samples with the highest addition of *Primula veris* L. flowers (10%), compared to the commercial wines without the flower addition. On the other hand, the lowest, four-fold reduction in anthocyanin content was noted for samples with a 2.5% addition of *Primula veris* L. The reason for the major reduction in anthocyanin content in the enhanced commercial wine samples was most likely their dilution with water contained in fresh *Primula veris* L. flowers, the emergence of conditions favorable for the deactivation/decomposition of pigments in red wines, and their high instability during wine sample storage at room temperature during the 14-day extraction. In this respect, regional wines demonstrated not only a higher stability, but also a high ability to extract anthocyanins from *Primula veris* L. The initial anthocyanin content in regional wine samples increased four times to the level of 1956.85 mg/L for a 10% addition of *Primula veris* L. flowers. In the case of a 2.5% addition of these flowers, the anthocyanin content in regional wine samples increased almost twice, compared to non-enhanced wines.

A high content of polyphenolic compounds of the flavonol group was found in both the regional and commercial red wine samples. The mean content of these substances is shown in Table 4. 

The regional wines were characterized by a flavonol content approximately 62% lower than the commercial wines, in which the content of these substances was on the level of 104.55 mg/L. At the lowest *Primula veris* L. flower addition (2.5%), the flavonols content in the regional wines increased almost 10-fold, while at the 10% *Primula veris* L. addition, the flavonols content was 1883.60 mg/L, which is almost 30 times higher than in wine samples without the addition of *Primula veris* L. flowers. In the commercial red wines, the flower content led to an increase in total flavonol content to the level of 366 mg/L, and for the 10% addition of *Primula veris* L. flowers, the content of these substances increased 14 times. An increase in kaempferol and quercetin was observed in the same samples, to the levels of 100.50 mg/L and 38.05 mg/L, respectively, while in the regional wines with a 10% *Primula veris* L. enhancement, kaempferol content was almost four times lower, and quercetin almost three times lower than in similar samples of enhanced commercial wines. The regional wines exhibited superior extraction abilities in relation to polyphenolic compounds from *Primula veris* L. flowers, but a lower content of health-crucial anti-oxidants and their derivatives. An analysis of the polyphenolic compound profile in the white wines, shown in Table 5, confirmed the excellent extraction abilities of both regional and commercial wines. 

For white wines, a five-fold increase in flavonol content was found in regional and wine samples, and an almost 25-fold increase in commercial wine samples at the lowest *Primula veris* L. flower addition of 2.5%. When comparing the results of the polyphenol profile analysis, the white wines can be considered more effective extractants than the test red wines. In the case of white wines, the mean flavonol content ranged from 92.50 mg/L in the regional wines to 64.35 mg/L in the commercial wines, and at the 10% addition of *Primula veris* L. flowers, the flavonol content was 2356.80 mg/L and 2404.30 mg/L, respectively, which exceeded the highest content of these substances in the red regional wines by approximately 27%. Qualitative analysis of the regional white wines with a 10% *Primula veris* L. flower enhancement demonstrated the highest kaempferol content (197.75 mg/L), and a high quercetin content (31.35 mg/L), while for the white commercial wines enhanced with the same amount of flowers, the content of these substances was eight times lower for kaempferol and 2.6 times lower for quercetin.

The assessment of the pro-health properties of wines enhanced with *Primula veris* L. flowers is also based on quantitative and qualitative assay of saponins in red and white wines, the results of which are shown in Table 6.

It was observed in all the test samples of enhanced wines that the substance present in the greatest quantities was Primula saponin I, particularly in the regional red wines. On the other hand, the lowest content was observed for Priverosaponin B 22-acetate, in particular in the commercial red wines enhanced with *Primula veris* L. According to multiple authors, this substance is a saponin characteristic of *Primula veris* L. roots [13]. The total saponin content in the regional red wine blends was 145.70 mg/L at the 2.5% addition of *Primula veris* L. flowers, while at the 10% flower addition, the saponin content was 438.35 mg/L. In the red commercial wines enhanced with the lowest *Primula veris* L. addition, on the other hand, the total saponin content was found to be 103.90 mg/L, while at the highest flower addition, the total content of these substances increased almost three-fold. In the case of the commercial white wines, the 2.5% cowslip flower addition enhanced the wine blends with 120.20 mg/L of saponins, while at the 10% flower addition, the total content of these substances was 270.80 mg/L. The regional white wines were characterized by an ability to extract saponins to solutions on a similar level to the regional red wines, i.e., 131.75 mg/L, while when enhanced with 10% *Primula veris* L., the saponin content increased to 299.65 mg/L.

Considering the varied extraction ability of the white and red wines in relation to the pro-health substances from *Primula veris* L., the impact of ethanol strength on extraction when ethanol was used to extract polyphenolic substances and saponins from *Primula veris* L. flowers that were added to regional and commercial wines. An analysis of the results, which is shown in Table 7, confirmed the adverse effect of ethanol strength on the extraction of polyphenolic compounds from *Primula veris* L. flowers. 

In samples with the lowest ethanol strength (40%), the highest mean flavonol content of 145.70 mg/L among all ethanol samples was achieved, while for the highest ethanol strength (95%), the value was reduced to 11.00 mg/L. The results of analysis of saponin content in ethanol blends also confirmed the statistically highly significant differences between the mean saponin content in ethanol solutions, with the relation being reverse to that in the case of flavonols. Ethanol strength had a beneficial impact on saponin extraction from cowslip flowers. In samples with the lowest ethanol strength (40%), a concentration of 168.50 mg/L was found, while at the highest ethanol strength, more than 2.5 times more saponins were found than for the lowest value. As in the case of wines enhanced with *Primula veris* L. flowers, Primula saponin I was extracted in the greatest amounts (117.90–317. 65 mg/L), while Priverosaponin B22 was present in the smallest quantities (16.25–33.00 mg/L).

Table 8 shows the results of statistical analysis of the influence of type of wine and the level of *Primula veris* L. addition on the profile of polyphenolic compounds and antioxidant properties, ethanol, and sugars content in enhanced wines. 

Statistical analysis demonstrated highly significant differences between anthocyanin content in the regional red wines, which were characterized by a four times higher content of these substances compared to the commercial wines. Furthermore, statistically highly significant differences between the total saponin content in the enhanced wine samples were confirmed. The red wines displayed a 25% higher saponin content as compared to the remaining wine samples in this respect. Additionally, the regional red wines also showed the highest antioxidant potential, as determined using the ferric reducing antioxidant power method (FRAP), DPPH, and 2,2-azino-bis(3-ethylbenzothiazoline-6-sulfonic acid) (ABTS), as well as total phenolic compounds (TPC) methods. Lower antioxidant ability assay values were found for white wines, particularly the regional ones, which were, on the other hand, characterized by the highest sugar content among all the test samples. The commercial white wines displayed the highest flavonol content. Cowslip enhancement resulted in statistically highly significant differences in each of the analyzed ingredients of white and red wines. The highest flower content of 10% markedly improved the content of anthocyanins in wines by approximately 80%, the flavonol content as compared to wines without cowslip addition by up to 28 times, and the saponin content almost 2.5 times as compared to the lowest *Primula veris* L. flower addition. Cowslip addition also resulted in a statistically high improvement of the anti-oxidant ability of the test wines, as determined using the FRAP and DPPH, and particularly ABTS methods, increasing this value more than 130 times. Wine enhancement with cowslip resulted in a reduction in alcohol content, and a slight increase in sugar content. However, this did not significantly affect wine flavor or its classification, and the wines remained in the dry wine segment. The content of residual sugar in dry wines ranges between 2 and 4 g/L [4], meaning that all the tested wines were classified as dry.

## 3. Discussion

Over 1000 mineral and organic compounds have been discovered in wines to date. The content of these compounds depends on the following factors: grapevine variety, climate and soil conditions, and grape ripeness [3].

Assessing the quality of wines enhanced with herbal plants that were analyzed in this paper is difficult due to the lack of literature data on this subject. There is also little data in the scientific literature concerning the quality of Polish wines. Analyzing reports on the relation between the pro-health quality of wines and their method of production, one can find information, for example, that red wines contain 10 times as much polyphenolic compounds as white wines [18]. This mainly results from their production method. The polyphenol content in wine depends mainly on the maceration process and its duration. If the grapes are pressed too quickly, a light-colored grape must forms, as a large group of polyphenolic compounds, e.g., anthocyanins, is located in grape skin. Polyphenolic compounds have a major influence on such properties as color, sharpness, and bitterness, and are therefore important indicators in grapevine cultivation and enology. A particularly important role in the quality of red grapes and wine is played by color anthocyanins, whose peculiar profile is the main indicator for classifying grape and wine varieties [19].

In general, anthocyanin content in wines produced from interspecies hybrid grapes (e.g., the tested regional wine Rondo) is higher and ranges from 400 to 700 mg/L, and in some cases may exceed 2000 mg/L [20,21]. In traditional wine production, anthocyanin concentration may change after a few days of fermentation, due to the absorption of certain anthocyanins by yeast cell walls, and precipitation as wine salts; therefore, the anthocyanin profile even in wines made of the same grapevine varieties may significantly differ [22]. One should mind that the grapevines cultivated in Poland are a relatively new material of poor quality [20,21,22,23,24,25]. Samoticha et al. [26] demonstrated that the highest concentration of anthocyanins occurred in the Regent and Rondo wines produced in Poland, but the content of their derivatives depends on the grape variety as well as the winemaking technique.

In this study, flavonol profiles concerning white wine characteristics contained significantly more derivatives than those reported in the studies that are available in the literature, as other authors very frequently mention only the major compounds, such as myricetin glycosides, quertecin, and to a lesser extent kaempferol [27]. According to the data shown by Makris et al. [28], the flavonol content in white wines produced from various grape varieties and in different regions may range from 2 to 7 mg/L. In this study, flavonol content was on average twice as high as in wines from other regions. In the case of red wines, flavonol content can range from 5 mg/L to 100 mg/L, according to literature data. It is believed that the flavonoids present in red wine are responsible for so called “French paradox”, which involves a low occurrence of cardiovascular diseases in populations with high red wine intakes [29].

The determination of ethyl alcohol content is of high importance in the winemaking industry. This parameter is used to control the fermentation process and for certification of alcoholic and non-alcoholic products. It is the main organic by-product of fermentation performed by yeast [30]. Most countries require stating ethanol content on labels, and moreover, wine price depends on the content of this compound in some cases [3]. The low ethanol content in Polish wines—9.7% to 11.9% *v*/*v* on average, and 12.5% *v*/*v* for the Rondo variety—has been observed by Tarko et al. [9,31]. On the other hand, in the Kapusta et al. [4] study, ethanol content was 12.44% *v*/*v* on average in red wines, and 12.30% *v*/*v* in white wines. The results concerning ethanol content in the test red and white wine samples in this study were similar to the literature data and consistent with the manufacturers’ information on the labels. The alcohol content was reduced in samples enhanced with *Primula veris* L.

Sugar content in grapes depends mainly on the variety and on fruit ripeness. It is also affected by fermentation microflora, as mentioned previously. In the Polish climate, the ripening process runs in a manner largely different than in warmer regions. It results in a significantly lower sugar content, which may also translate to a lower ethanol content [32]. The main sugars present in grapes are the monosaccharides glucose and fructose, which are key for the growth and development of yeast. When the fermentation process is complete, so-called residue sugar remains, which is made up of pentose sugars (arabinose, rhamnose, xylose) and a small quantity of unfermented fructose and glucose [3]. 

The antioxidant properties of wines vary greatly depending on the grapevine species as well as environmental and geographical factors. They result in particular from the content of biologically active compounds, such as phenolic compounds, vitamins, and enzymes. As has been noted previously, red wines are a richer source of polyphenols and display relatively higher antioxidant properties in comparison to white wines. In studies performed by other researchers, it has been demonstrated that the varieties analyzed in this study, i.e., Rondo, are particularly rich in polyphenolic compounds and display a high antioxidant activity [5]. In the study of Kapusta et al. [4], who analyzed wine samples from Subcarpathian vineyards, antioxidant activity measured by the FRAP method ranged from 2 mmolTE/L (Cabernet Cortis variety) to 0.05 mmolTE/L (Regent variety); the Regent variety also contained the most polyphenolic compounds as assayed using LC/MS (1860 mg/L). The Rondo wines were also characterized by a very high content of polyphenolic compounds in the Socha et al. [33] study, 996 and 1669 mg GA/L, respectively. Furthermore, Socha et al. [33] observed that wines from the southern regions of Poland have a similar content of polyphenolic compounds to wines from regions with continental climates, but lower than red wines from warm regions, such as Greece, Portugal, or Italy. 

A study conducted in recent years in Southeastern European countries demonstrated a rich biovariety and exceptional vitality of traditional plant knowledge in that region [34,35]. Familiarity with cowslip (*Primula veris* L.) in that part of Europe is not limited merely to applications in traditional medicine; it is also used as an additive in food [36]. This traditional plant product is widely used in several Central European countries in herbal preparations and pharmaceutical formulations, and its biological and pharmacological activity has been confirmed both in scientific and medical literature. Cowslip (*Primula veris* L., syn. *P. officinalis* Hill) and oxlip (*Primula elatior* (L.) Hill) are small, long-lived perennials from the family Primulaceae, growing wild in Europe and Asia [37]. Both species have a long history of medicinal use. In the current issue of the European Pharmacopoeia, they are listed as a source of *Primula* root, from which bioactive substances for pharmacological applications are acquired [38]. However, in the British Herbal Pharmacopoeia [39], as well as the Pharmacopee Francaise [40], only *Primula veris* L. is listed as a source of raw material for the production of pharmaceutical and pro-health preparations with mucoactive, anti-inflammatory, diuretic, antimicrobial, antifungal, and sedative effects [16,41,42].

The main ingredient used by the pharmaceutical industry from *Primula veris* L. is saponins, whose content in the plant’s roots may reach 5–10%. Saponins from *Primula veris* L. are triterpenoid glycosides with the oleanane ring system linked to carbohydrate moieties [43].

However, the authors of other studies note difficulties in the precise qualitative determination of the content of saponins extracted from *Primula veris* L. roots, which is most likely a consequence of a lack of chromophore groups, impairing UV detection. Due to the great genetic variation and the significance of climate, soil, and geography conditions in determining the content of bioactive substances in *Primula veris* L., a clear determination of the profile of pro-health properties for cowslip is extremely difficult [44,45]. Many authors focus their attention primarily on the roots of *Primula veris* L., considering the potential concentration of bioactive substances in these morphological parts [46]. Other researchers, basing on the latest analytical techniques, question the profiles of bioactive compounds from Primula veris L., as determined previously [14]. 

According to own research, *Primula veris* L. flowers are equally valuable as an object of research, as they are rich in bioactive substances, and their extraction does not require complex preparative techniques. 

In the latest study performed by Apel et al. [47], who used high performance liquid chromatography/diode array detector/mass spectrometry (HPLC-DAD-MS) to study the methanol extracts from the petals of three Primula varieties, including *Primula veris* L., a broad spectrum of flavonoids and their glycosylated or methylated derivatives has been shown, which is consistent with the results obtained in our study. Furthermore, these researchers demonstrated that extracts from the leaves of three *Primula* samples were characterized by highly similar flavonoid and saponin profiles to the leaves; however, individual compounds were present in different relative quantities. The root extracts of all three Primula samples provided identical compound profiles, which consisted of triterpenoid saponins, but were devoid of the flavonoids and anthocyanins detected in the above-ground plant organs. Among saponins, the highest concentration in the test extracts was observed for Primula saponin I, which our study has confirmed as well. According to the study by Müller et al. [17], who tested methanol extracts obtained from dried roots and flowers of *Primula veris* L., the saponin dominant in the flowers is Primula saponin I, whose content can range from 0.22% to 0.28%. On the other hand, priverosaponin B-22-acetate has been identified only in root extracts, which according to the authors confirms the earlier reports concerning the saponin profile in various parts of the plant and contradicts the results obtained in our study [17]. Considering the above reports, it can be concluded that the content of biologically active substances in *Primula veris* L. has not been unambiguously determined and still leaves hope for new applications for the pro-health substances contained in *Primula veris* L. In the Committee on Herbal Medicinal Products (HMPC) report [41], information can be found about unspecified saponin of the Primula root, which administered parenterally in a dose of 40 mg/kg inhibited the growth of Walker carcinoma in rats. 

Herbal medicinal products appear to be very promising as they have a noticeable therapeutic effect and tend to be more harmless in comparison to the most synthesized medications.

The study by Latypova et al. [48] focused particular attention on the identification of the raw material composition of the *Primula veris* L. The object of the analysis was a solid herbal extract and its effects on the myocardial contractile function in animals with experimental CHF (chronic heart failure). The authors of the study found that a solid herbal extract obtained from *Primula veris* L. contained flavonoid aglycons, flavonoid glycosides, and polymethoxylated flavonoids. It was further shown that the studied herbal agent at a dose of 30 mg/kg had a cardioprotective effect, as evidenced by a smaller number of animals deaths, the lower level of CHF plasma markers, a higher increase in myocardial contraction, and relaxation rates as compared to the control group [48].

Junqing et al. [49] found that herbs with a high percentage of phenolic constituents stimulate the synthesis of vascular endothelial growth factor and blood vessel density, and exert cardioprotection through promoting angiogenesis in the animal models of myocardial infarction [49,50].

The above effects of the studied herbal medicinal product from *Primula veris* L. are likely to be of key importance for undertaking further studies in this area, with the aim of providing cardioprotective benefits. The cardioprotective effects of wine has so far been attributed to resveratrol, which is a substance that is contained mainly in red wines, and in minimal quantities in white wines. Thanks to wine enrichment in triterpenoid saponins and in polyphenolic compounds from *Primula veris* L. flowers, which are effectively extracted to wine under mild conditions, both white and red wines can constitute a highly pro-health component of diets, which is valuable in preventing chronic heart failure.

## 4. Materials and Methods

### 4.1. Plant Material and Experimental Conditions

The test material (named Regional) were Polish, regional Rondo dry red wines (n = 3) from the “Wierzchowina” vineyard (marked W on Figure 1), located in Łęki Dolne, near Pilzno, at the base of the Carpathians in Poland, and the Rezeda Polish white wines (n = 3) produced at the regional vineyard “Jasiel” in Jaslo (marked J on Figure 1) in the Subcarpathian region, Poland. Rondo is an interspecies hybrid from Germany that is characterized by good resistance to frost (below −20°C) as well as pests, popular, and recommended for red wine production. Rezeda, on the other hand, is a mixture of the Muskaris, Siegerrebe, and Solaris varieties. The wine samples were stored in the dark, at a temperature of 4 °C, and opened directly before preparing the herbal blends. 

For the purpose of comparing selected properties of the Subcarpathian regional products, red *Cabernet Sauvignon* (n = 3) and white *Chardonnay* (n = 3) wines, commercially available under the Carlo Rossi brand, were selected as control samples (named Commercial). Furthermore, fresh *Primula veris L* inflorescences were used in the tests. The test material were inflorescences of the cowslip (*Primula veris* L.) plant, which were collected at the beginning of April 2018 from environmentally friendly herb cultures in the Subcarpathian region, in Jasielski powiat. Washed and dried inflorescences were divided into portions that formed the charge for the wine blends in the amounts of 2.5%, 5%, and 10%, which were then immersed in white and red wine (regional and commercial) in broad-necked glass bottles. Subsequently, the samples were capped and left at room temperature (25 °C) in an air-conditioned room for a period of 14 days. The wine blends were stirred every day by vigorously turning the closed bottles upside down 3 times. Additionally, alcohol solvents of different alcohol strengths were prepared (40%, 70%, and 96%) using ethanol and distilled water, together with equal additions of *Primula veris* L. in the amount of 5% for this purpose. The samples were prepared in closed glass bottles and handled identically as the wine blends. Once the 14-day free extraction was complete, the wine samples were filtered and prepared directly for further analyses. Before analyzing the bioactive profiles, the wine and ethanol samples were filtered through paper filters at vacuum. Clear wine and alcohol solutions were applied to conditioned column beds of a Thermo Scientific™ SPE 16- or 24-Port SPE Vacuum Manifolds filtration system (C_18_ 500mg bed). Polyphenolic compounds were eluted from the columns using ethanol directly to round-bottom flasks, and concentrated at a temperature of 40 °C in a Hei-VAP Precision rotary vacuum evaporator manufactured by Heidolph (Germany), until complete evaporation of the solvent. The wine and alcohol extracts in round-bottom flasks were dissolved in methanol and filtered using PTFE filters with 0.45-µm pore diameter immediately before the chromatographic analysis.

### 4.2. Determination of Polyphenolic Compounds

The analysis was performed according to the method described by Kapusta et al. [51]. Polyphenolic compounds were analyzed using ultra-performance reverse-phase liquid chromatography (UPLC)-PDA-MS/MS Waters ACQUITY system (Waters, Milford, MA, USA), consisting of a binary pump manager, sample manager, column manager, photodiode array (PDA) detector, and tandem quadrupole mass spectrometer (TQD) with electrospray ionization (ESI). The separation was carried out using a BEH C18 column (100 mm × 2.1 mm i.d., 1.7 μm, Waters) kept at 50 °C. For the anthocyanins investigation, the following solvent system: mobile phase A (2% formic acid in water *v*/*v*) and mobile phase B (2% formic acid in 40% ACN in water *v*/*v*) were applied. For other polyphenolic compounds, a lower concentration of formic acid was used (0.1% *v*/*v*). The gradient program was set as follows: 0 min 5% B, from 0 to 8 min linear to 100% B, and from 8 to 9.5 min for washing and back to initial conditions. The injection volume of the samples was 5 μL (partial loop with needle overfill), and the flow rate was 0.35 mL/min. The following parameters were used for TQD: capillary voltage, 3.5 kV; con voltage, 30 V in positive and negative mode; the source was kept at 120 °C and the desolvation temperature was 350 °C; con gas flow, 100 L/h; and desolvation gas flow, 800 L/h. Argon was used as the collision gas at a flow rate of 0.3 mL/min. The polyphenolic detection and identification were based on a specific PDA spectra, mass-to-charge ratio and fragment ions obtained after collision-induced dissociation (CID). Before injection, wine samples were filtered through a 0.45-μm pore-size membrane filter (Merck Millipore) and injected directly into a chromatographic column. Quantification was achieved by the injection of solutions of known concentrations ranging from 0.05 to 5 mg/mL (R^2^ ≤ 0.9998) of phenolic compounds as standards. All determinations were performed in triplicate and expressed as mg/L. Waters MassLynx software v.4.1 was used for data acquisition and processing. 

### 4.3. Determination of Saponins

Saponins were quantitated using ultra-performance reverse-phase liquid chromatography (UPLC) coupled with ESI-MS/MS described by Mroczek et al. [52]. Analyses were carried out using an Acquity ultra-performance liquid chromatograph (Waters) coupled with an Acquity TQD tandem quadrupole mass spectrometer with an ESI source. The separation was undertaken using a 100 × 2.1 mm i.d., 1.7 μm, Acquity UPLC BEH C_18_ column. A mobile phase consisting of 0.1% formic acid in acetonitrile (B) and 0.1% formic acid in water (A) was used for the separation. The gradient elution was linear from 25% to 60% B, over 0–6 min; isocratic at 60% B, over 6–6.5 min; linear from 60% to 25% B, over 6.5–6.6 min; and isocratic at 25% B, over 6.6–7 min. The column was maintained at 50 °C at a constant flow rate of 0.3 mL/min. The sample injection volume was 5 μL. The following instrumental parameters were used for the ESI-MS analysis of saponins: capillary voltage, 3.5 kV; cone voltage, 80 V; desolvatation gas, nitrogen 800 L/h; cone gas, nitrogen 100 L/h; source temperature, 120 °C; desolatation temperature, 350 °C; and dwell time, 0.05 s. The detection mode was SIR in negative ion mode. Quantitation was based on external standardization by employing the calibration curve of oleanolic acid in the range of 80–560 ng/mL. The quantitative analyses were based on the peak area calculated from the selected ion chromatograms of the corresponding [M − H]^−^ ion, and saponins were identified through a comparison of their retention times and ion mass. Microsoft Excel 2000 was used for statistical analysis.

### 4.4. Analysis of Sugar Profiles

The sugar characteristics of wines were assessed by HPLC according to the method described by Dżugan et al. [53] with our own modifications. The sugar profile of samples was analyzed by high-performance liquid chromatography using a Thermo Dionex Ultimate 3000 with a charged aerosol detector (CAD) and ultraviolet-diode array detector (UV/DAD) (Thermo Fisher Scientific Inc., Waltham, MA, USA). Chromatographic separation was performed using a Shodex Asahipak NH2P-504E column (C18, 250 nm × 4.6 nm) and acetonitrile:water (78:22, *v*/*v*) as the mobile phase (1 mL/min). The injection volume was 10 μL, the temperature was set at 35 °C, and the analysis time was 30 min. Before injection, the samples were filtered using PTFE filters (0.22 μm). Sugars were identified on the basis of their retention times. The average yield for sugars in wines was 90–94%. The precision of this analytical method was confirmed by repeating the standard and all sample injections three times. Chromatographic system stability was controlled in four-hour intervals by injecting selected standard solutions of known concentration.

### 4.5. Analysis of Antioxidant Activity

Antioxidant activity was measured by two different method: The ferric reducing antioxidant power method (FRAP) and radical scavenging activity (DPPH). Ferric reducing antioxidant power (FRAP) was performed according to Benzie and Strain [54]. The results were expressed as mmol of Trolox equivalents per 1 L of tested samples (mmol TE /L), using a calibration curve plotted for Trolox solution in a concentration range of 0.05–0.30 µmol/mL (R^2^ = 0.9967). Scavenging activity was determined according to the elimination of DPPH radicals [55].

Antiradical activity was carried out using the ABTS decolorization assay described previously by Re et al. [56]. Determinations were performed in triplicate, and results were expressed as mmol of ascorbic acid equivalent per L. The determination of ABTS, DPPH, and FRAP methods was performed using a Spectrophotometer UV VIS UV6000 (Shanghai Metash Instruments Co., Shanghai, China).

### 4.6. Analysis of Total Phenolic Compounds (TPC)

The content of total phenolic compounds (TPC) was investigated by Folin–Cocialteu’s method as described by Stratil et al. [57]. The results were expressed as 1 mg of gallic acid equivalents per l (mg GAE/L) of tested samples, using a calibration curve plotted for GAE solution in a concentration range of 25–250 µg/mL (R^2^ = 0.997). The measurements were performed using a Spectrophotometer UV VIS UV6000 (Shanghai Metash Instruments Co., Shanghai, China).

### 4.7. Determination of Ethanol Content

The ethanol content in the wine samples was determined using the colorimetric method according to the methodology developed by Gonchar et al. [58], using the Alkotest analytical system, containing alcohol oxidase, peroxidase, chromogen, and buffering components with some modifications. The reaction was interrupted when the samples turned light blue by adding 100 μL of 0.8 M hydrochloric acid. Absorbance was measured at the 450 nm wavelength against a blank sample. The alcohol concentration results were read using a calibration chart, and then calculated in a manner similar to that shown in the Gonchar et al. [58] paper, and expressed in % *v*/*v*.

### 4.8. Chemicals and Reagents

The determinations were performed using analytical grade reagents intended for liquid chromatography: hydrochloric acid, formic acid, ethanol by Sigma-Aldrich; methanol by J.T. Baker Malinckrodt Baker B.V. Holland. Acetonitryl CHROMASOLV^®^ gradient grade, ≥99.9% by Sigma-Aldrich. Sugar analytical standards by BioXtra, ≥99% HPLC grade, were obtained from Sigma-Aldrich. Deionized water was obtained using a HLP 5P deionizer manufactured by Hydrolab Polska. Oleanolic acid standard, 2,2-azino-bis(3-ethylbenzothiazoline-6-sulfonic acid) (ABTS) diammonium salt, 6-hydroxy-2,5,7,8-tetramethylchroman-2-carboxylic acid (Trolox), 2,4,6-tri(2-pyridyl)-s-triazine (TPTZ), and 2,2-diphenyl-1- picrylhydrazyl (DPPH) were purchased from Sigma-Aldrich (Steinheim, Germany).

### 4.9. Statistical Analysis 

All the analyses were made in three independent replications for each sample. The results were presented as an arithmetic mean ± standard deviation (SD). The findings were subjected to statistical analyses with the use of Statistica 13.1 software (StatSoft, Inc., Tulsa, OK, USA). The significant differences between the mean values were obtained by a two-way analysis of variance (ANOVA) followed by Duncan’s multiple range test.

## 5. Conclusions

As our study indicates, ethanol content has a significant impact on the extraction of health-beneficial polyphenolic compounds, reducing its effectiveness, while it advantageously affects saponin extraction from *Primula veris* L. flowers, increasing their concentration at higher ethanol strength levels. From the perspective of pro-health and the cardioprotective use of cowslip extracts, it is much more beneficial to prepare them based on white or red wines, which are products of low alcohol content, as it does not impair extraction, and additionally enriches the blends with the particular profile of the pro-health compounds contained therein.

Considering the above reports and the wealth of polyphenolic compounds contained in the analyzed red and white wines enhanced with *Primula veris* L. flowers, further action should be taken to clearly determine the ranges of fortification with cowslip extracts, as well as develop pro-health preventive practices that make use of wines enhanced with *Primula veris* L. flowers.

## Figures and Tables

**Figure 1 molecules-24-04074-f001:**
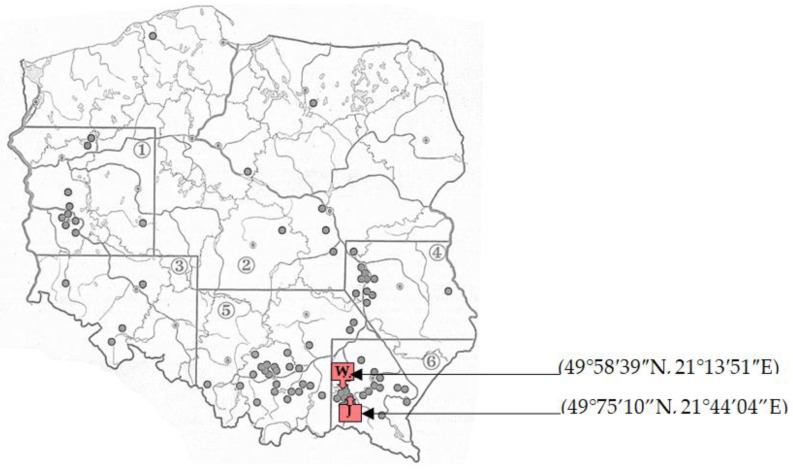
Wine Regions in Poland with some of the major vineyards highlighted (points). 1—Zielona Góra; 2—Central and Northern Poland; 3—Lower Silesia; 4—Lesser Poland Gorge of the Vistula (with Lublin Voivodeship); 5—Lesser Poland (with Sląskie and Świętokrzyskie Voivodships); 6—Podkarpacie.

**Table 1 molecules-24-04074-t001:** Individual anthocyanins identified by ultra-performance reverse-phase liquid chromatography (UPLC)-PDA-MS/MS in red wine.

No	Compound	RT	[M − H]^+^	Fragment Ions	Absorbance Maxima
		(min)	(*m*/*z*)	(*m*/*z*)	(nm)
1	Delphinidin 3-*O*-glucoside-5-*O*-glucoside	2.04	627	465, 303	277, 525
2	Cyanidin 3-*O*-glucoside-5-*O*-glucoside	2.19	611	449, 287	280, 516
3	Delphinidin 3-*O*-glucoside	2.38	465	303	280, 523
4	Petunidin 3-*O*-glucoside-5-*O*-glucoside	2.53	641	479, 317	277, 531
5	Peonidin 3-*O*-glucoside-5-*O*-glucoside	2.67	625	463, 301	278, 513
6	Malvidin 3-*O*-glucoside-5-*O*-glucoside	2.72	655	493, 331	275, 524
7	Cyanidin 3-*O*-glucoside	2.74	449	287	279, 515
8	Petunidin 3-*O*-glucoside	2.92	479	317	277, 526
9	Peonidin 3-*O*-glucoside	3.31	463	301	279, 515
10	Malvidin 3-*O*-glucoside	3.43	493	331	278, 530
11	Delphinidin 3-*O*-(6″-*O*-acetyl)-glucoside	3.53	507	465, 303	280, 528
12	Cyanidin 3-*O*-(6″-*O*-acetyl)-glucoside	3.95	491	449, 287	283, 522
13	Petunidin 3-*O*-(6″-*O*-acetyl)-glucoside	4.1	521	317	280, 530
14	Petunidin 3-*O*-(6″-*O*-acetyl)-glucoside-5-*O*-glucoside	4.28	787	625, 479, 317	280, 530
15	Delphinidin 3-*O*-(6″-*O*-coumaryl)-glucoside	4.47	611	303	279, 530
16	Malvidin 3-*O*-(6″-*O*-acetyl)-glucoside	4.62	535	331	280, 521
17	Malvidin 3-*O*-(6″-*O*-coumaryl)-glucoside-5-*O*-glucoside	4.67	801	639, 493, 331	280, 530
18	Peonidin 3-*O*-(6″-*O*-coumaryl)-glucoside-5-*O*-glucoside	4.68	771	609, 463, 301	279, 523
19	Peonidin 3-*O*-(6″-*O*-acetyl)-glucoside	4.85	505	463, 301	277, 535
20	Cyanidin 3-*O*-(6″-*O*-coumaryl)-glucoside	4.93	595	287	283, 522
21	Petunidin 3-*O*-(6″-*O*-coumaryl)-glucoside	4.98	625	317	280, 531
22	Delphinidin 3-*O*-(6″-caffeoyl)-glucoside	5.35	627	465, 303	280, 528
23	Peonidin 3-*O*-(6″-*O*-coumaryl)-glucoside	5.39	609	301	279, 523
24	Malvidin 3-*O*-(6″-*O*-coumaryl)-glucoside	5.44	639	331	280, 521

**Table 2 molecules-24-04074-t002:** Individual flavonols and saponins identified by (UPLC)-PDA-MS/MS in wine and ethanol.

No	Compound	Rt	[M − H]^+^	Fragment Ions	Absorbance Maxima
		(min.)	(*m*/*z*)	(*m*/*z*)	(nm)
1	Quercetin 3-*O*-(6″-*O*-rhamnosyl)-glucoside 7-*O*-rhamnoside	4.13	755	609, 301	254, 353
2	Metyl-miricetin-3-*O*-rutinoside	4.28	639	493, 331, 317	276, 339
3	Metyl-miricetin-*O*-glucoside	4.33	493	331, 317	276, 338
4	Kaempferol 3-O-rutinoside 7-*O*-rhamnoside	4.50	739	593, 285	264, 347
5	Metyl-quercetin 3-*O*-(6″-*O*-rhamnosyl)-glucoside 7-*O*-rhamnoside	4.58	769	755, 609, 301	253, 354
6	Quercetin 3-*O*-rutinoside	4.65	609	301	255, 354
7	Quercetin 3-*O*-glucoside	4.79	463	301	255, 355
8	Kaempferol 3-*O*-rutinoside	4.84	593	285	262, 352
9	Kaempferol 3-*O*-rhamnosyl-glucoside isomer I	4.96	593	285	264, 348
10	Kaempferol 3-*O*-rhamnosyl-glucoside isomer II	5.15	593	285	264, 347
11	Metyl-quercetin 3-*O*-rutinoside	5.21	623	609, 301	253, 353
12	Metyl-quercetin 3-*O*-rhamnosyl-glucoside	5.28	623	609, 301	255, 354
13	Undefined kaempferol derivative	5.48	507	493, 285	269, 355
14	Quercetin 3-*O*-glucuronide	5.52	477	301	255, 354
15	Kaempferol 3-*O*-pentosyl-pentoside	5.87	549	285	269, 355
16	Kaempferol	6.94	285	-	264, 367
17	Quercetin	7.02	301	-	255, 355
18	Isorhamnetin	8.35	315	-	255, 355
1	Primula saponin II	4.19	1236	924, 465	
2	Primula saponin I	4.31	1104	924, 465, 447, 246	
3	Priverosaponin B 22-acetate	4.53	1162	982, 465	

**Table 3 molecules-24-04074-t003:** The content of anthocyanins in red wines enriched with *Primula veris* L. (mg/L).

Red Wine	Regional				Commercial			
*Primula veris* L. (%)	0	2.5	5	10	0	2.5	5	10
**Anthocyanins**								
**1**	0.11 ± 0.05	22.15 ± 0.50	23.45 ± 0.35	30.70 ± 0.85	tr	tr	tr	tr
**2**	0.06 ± 0.11	4.95 ± 0.05	5.65 ± 0.15	5.72 ± 0.25	tr	tr	tr	tr
**3**	1.44 ± 0.30	6.85 ± 0.00	7.72 ± 0.05	9.65 ± 0.61	tr	tr	tr	tr
**4**	0.43 ± 0.35	30.95 ± 0.30	35.61 ± 1.30	44.45 ± 0.92	tr	tr	tr	tr
**5**	0.10 ± 0.00	21.50 ± 0.70	28.15 ± 0.35	31.71 ± 3.30	15.02 ± 0.25	14.45 ± 0.15	tr	tr
**6**	6.22 ± 0.20	19.31 ± 0.15	42.55 ± 1.87	38.22 ± 15.95	14.45 ± 0.20	14.02 ± 0.35	13.42 ± 1.11	tr
**7**	3.55 ± 0.65	5.65 ± 0.20	6.25 ± 0.25	7.50 ± 0.30	4.53 ± 0.15	tr	tr	tr
**8**	6.16 ± 0.00	7.77 ± 0.55	16.55 ± 0.11	18.81 ± 1.65	6.61 ± 0.20	tr	5.30 ± 0.85	tr
**9**	0.52 ± 2.05	18.11 ± 0.20	20.75 ± 1.35	25.62 ± 0.65	15.70 ± 0.20	15.01 ± 0.40	39.7 ± 2.01	tr
**10**	0.52 ± 0.10	224.65 ± 8.31	306.70 ± 14.95	343.90 ± 24.31	52.04 ± 0.20	3.65 ± 0.05	4.42 ± 0.52	9.4 ± 0.25
**11**	2.61 ± 0.70	24.50 ± 0.30	27.22 ± 0.10	33.35 ± 0.92	23.12 ± 0.82	tr	tr	tr
**12**	1.20 ± 0.01	7.15 ± 0.55	8.75 ± 0.31	9.85 ± 0.55	6.92 ± 0.05	5.85 ± 0.05	5.21 ± 0.95	tr
**13**	0.81 ± 0.15	tr	tr	tr	4.85 ± 0.01	tr	tr	tr
**14**	6.45 ± 0.05	10.75 ± 0.10	12.75 ± 0.35	14.65 ± 2.55	6.65 ± 0.02	6.25 ± 0.12	tr	tr
**15**	16.35 ± 0.00	26.10 ± 1.11	33.05 ± 0.85	38.02 ± 1.35	24.10 ± 0.17	52.15 ± 0.75	19.55 ± 3.22	tr
**16**	1.48 ± 0.36	17.15 ± 0.00	21.21 ± 0.40	23.90 ± 0.35	19.11 ± 0.91	15.35 ± 0.25	14.34 ± 1.05	15.45 ± 0.65
**17**	45.53 ± 3.85	21.15 ± 0.35	68.63 ± 0.92	76.91 ± 1.11	50.20 ± 2.70	51.45 ± 0.65	5.23 ± 0.20	14.95 ± 0.05
**18**	13.05 ± 0.15	92.90 ± 3.21	171.80 ± 9.55	183.35 ± 2.55	111.90 ± 0.75	5.82 ± 0.30	0.35 ± 0.01	2.25 ± 0.01
**19**	76.30 ± 2.40	21.21 ± 0.30	24.85 ± 0.25	37.42 ± 0.53	15.12 ± 0.15	tr	tr	tr
**20**	27.20 ± 1.35	112.35 ± 3.35	179.92 ± 2.85	275.60 ± 3.85	7.65 ± 0.12	12.03 ± 3.50	11.02 ± 1.82	12.85 ± 0.25
**21**	13.45 ± 0.05	40.91 ± 1.35	104.65 ± 3.75	162.55 ± 2.30	82.74 ± 1.50	68.95 ± 0.55	7.15 ± 1.15	29.8 ± 0.60
**22**	13.12 ± 0.05	22.25 ± 0.30	24.10 ± 0.15	36.65 ± 0.50	tr	tr	tr	tr
**23**	127.85 ± 5.75	19.13 ± 0.50	25.95 ± 0.70	40.02 ± 0.55	22.95 ± 0.32	16.11 ± 0.30	14.92 ± 1.30	15.35 ± 0.05
**24**	106.5 ± 4.80	134.03 ± 1.25	329.73 ± 5.55	493.55 ± 6.91	190.22 ± 0.35	29.35 ± 0.61	20.41 ± 2.05	11.15 ± 0.35
**Total**	**492.25 ± 3.80**	**918.00 ± 1.46**	**1525.75 ± 2.31**	**1956.85 ± 5.48**	**682.00 ± 2.53**	**282.60 ± 0.17**	**120.80 ± 3.48**	**111.15 ± 2.05**

mean values ± SD (n = 3). tr – traces under LOD (LOD - limit of detection).

**Table 4 molecules-24-04074-t004:** The content of flavonols in red wines enriched with Primula veris L. (mg/L).

Red Wine	Regional				Commercial			
*Primula veris* L. (%)	0	2.5	5	10	0	2.5	5	10
**Flavonols**								
**1**	tr	8.15 ± 0.00	14.80 ± 0.00	28.60 ± 0.00	4.95 ± 0.22	1.95 ± 0.00	15.65 ± 0.05	22.65 ± 0.45
**2**	tr	tr	tr	41.82 ± 0.55	tr	3.30 ± 0.11	24.05 ± 0.9	23.30 ± 0.00
**3**	tr	tr	tr	14.01 ± 0.15	tr	1.55 ± 0.11	10.95 ± 0.6	8.35 ± 0.05
**4**	tr	5.52 ± 0.25	10.92 ± 0.75	16.70 ± 0.55	18.70 ± 0.75	0.91 ± 0.05	5.45 ± 0.15	10.75 ± 0.15
**5**	tr	19.21 ± 0.65	39.30 ± 0.35	59.15 ± 0.95	9.62 ± 0.55	4.72 ± 0.00	29.65 ± 0.75	41.31 ± 1.71
**6**	1.3 ± 0.05	141.10 ± 8.25	203.51 ± 6.25	290.63 ± 4.85	7.35 ± 0.15	27.41 ± 1.50	138.71 ± 2.75	188.55 ± 4.62
**7**	tr	2.14 ± 0.31	5.62 ± 0.25	9.21 ± 0.22	7.11 ± 0.00	2.65 ± 0.30	6.82 ± 0.10	11.95 ± 0.45
**8**	1.5 ± 0.00	7.42 ± 0.05	12.45 ± 0.45	27.45 ± 0.10	16.51 ± 0.21	6.72 ± 0.50	10.50 ± 0.61	28.42 ± 0.75
**9**	0.95 ± 0.05	6.95 ± 0.05	13.00 ± 0.05	21.75 ± 1.41	16.13 ± 0.20	3.33 ± 0.05	11.45 ± 0.10	9.40 ± 0.02
**10**	0.45 ± 0.00	36.81 ± 0.05	65.02 ± 0.55	118.11 ± 0.15	3.45 ± 0.10	11.35 ± 0.25	48.55 ± 0.75	53.95 ± 0.15
**11**	0.95 ± 0.05	49.60 ± 0.23	87.05 ± 0.85	145.72 ± 2.62	5.25 ± 0.05	15.85 ± 0.05	64.85 ± 2.35	83.82 ± 0.30
**12**	4.5 ± 0.05	282.42 ± 1.25	520.05 ± 0.05	892.30 ± 0.50	tr	89.85 ± 3.25	359.21 ± 2.40	469.71 ± 8.20
**13**	tr	14.12 ± 0.2	14.35 ± 0.10	51.21 ± 0.35	tr	7.85 ± 0.11	21.61 ± 0.60	37.10 ± 0.00
**14**	2.35 ± 0.01	14.30 ± 0.3	21.75 ± 0.05	48.75 ± 0.15	tr	21.75 ± 0.95	45.25 ± 0.75	131.55 ± 0.40
**15**	16.25 ± 1.01	4.75 ± 0.15	6.75 ± 0.11	18.85 ± 0.00	tr	2.20 ± 0.10	6.72 ± 0.11	13.02 ± 0.05
**16**	tr	7.55 ± 0.05	8.85 ± 0.01	29.65 ± 0.25	4.42 ± 0.10	55.25 ± 0.40	78.65 ± 4.0	100.52 ± 1.82
**17**	5.6 ± 0.55	1.23 ± 0.05	3.52 ± 0.32	13.15 ± 0.20	8.05 ± 0.15	17.71 ± 1.15	30.45 ± 0.51	38.05 ± 0.50
**18**	29.25 ± 2.04	11.85 ± 0.05	13.75 ± 0.00	56.62 ± 0.05	3.21 ± 0.05	92.25 ± 2.45	162.45 ± 6.73	136.83 ± 1.65
**Total**	**63.55 ± 4.01**	**613.00 ± 2.25**	**1040.55 ± 5.30**	**1883.61 ± 4.70**	**104.55 ± 4.14**	**366.50 ± 4.68**	**1070.90 ± 4.21**	**1409.05 ± 3.09**

mean values ± SD (n = 3). tr – traces under LOD.

**Table 5 molecules-24-04074-t005:** The content of flavonols in white wines enriched with *Primula veris* L. (mg/L).

White Wine	Regional				Commercial			
*Primula veris* L. (%)	0	2.5	5	10	0	2.5	5	10
**Flavonols**								
**1**	tr	3.02 ± 0.00	5.65 ± 0.01	16.95 ± 0.12	tr	19.55 ± 0.50	24.21 ± 0.05	29.82 ± 0.50
**2**	tr	tr	tr	39.21 ± 0.50	0.42 ± 0.02	52.05 ± 4.05	65.41 ± 1.55	78.65 ± 1.50
**3**	tr	tr	tr	22.75 ± 0.65	tr	32.21 ± 0.45	30.60 ± 0.05	36.45 ± 0.25
**4**	tr	5.01 ± 0.31	5.25 ± 0.10	9.51 ± 1.02	tr	10.12 ± 0.11	12.10 ± 0.11	12.75 ± 0.45
**5**	tr	10.70 ± 0.25	19.12 ± 0.22	33.62 ± 0.11	0.45 ± 0.02	43.25 ± 0.45	45.21 ± 0.45	54.92 ± 1.02
**6**	39.50 ± 0.05	13.17 ± 0.85	85.60 ± 3.35	194.45 ± 10.75	3.92 ± 0.11	239.50 ± 4.21	365.15 ± 3.00	411.91 ± 1.25
**7**	4.35 ± 0.15	2.33 ± 0.01	4.72 ± 0.12	8.50 ± 0.21	4.71 ± 0.20	13.72 ± 0.15	8.55 ± 0.01	10.85 ± 0.01
**8**	16.22 ± 0.65	7.05 ± 0.25	28.71 ± 0.12	58.85 ± 2.75	17.10 ± 0.25	55.85 ± 0.05	75.02 ± 3.25	85.22 ± 0.15
**9**	4.45 ± 0.05	2.95 ± 0.00	9.83 ± 0.05	20.03 ± 0.22	4.61 ± 0.10	14.25 ± 0.05	15.85 ± 1.15	16.45 ± 0.00
**10**	1.55 ± 0.1	26.6 ± 0.00	36.80 ± 1.45	83.45 ± 1.70	2.12 ± 0.02	83.02 ± 0.10	136.31 ± 2.05	152.70 ± 0.95
**11**	1.60 ± 0.00	27.65 ± 0.05	105.62 ± 1.62	15.51 ± 0.15	2.21 ± 0.00	115.05 ± 6.01	126.32 ± 0.55	151.12 ± 2.52
**12**	1.92 ± 0.15	105.81 ± 0.40	527.01 ± 0.35	890.85 ± 2.35	11.95 ± 0.25	717.75 ± 4.8	1044.75 ± 5.15	1086.85 ± 2.05
**13**	10.65 ± 0.45	23.42 ± 0.32	48.05 ± 0.50	97.75 ± 1.05	1.15 ± 0.02	54.35 ± 2.21	69.20 ± 0.05	79.70 ± 1.35
**14**	0.45 ± 0.00	28.35 ± 0.05	81.65 ± 0.00	126.72 ± 1.05	3.62 ± 0.05	62.45 ± 5.45	75.25 ± 4.05	80.55 ± 1.81
**15**	1.95 ± 0.11	1.75 ± 0.00	5.55 ± 0.05	9.70 ± 0.05	1.70 ± 0.00	10.95 ± 0.42	20.04 ± 0.15	20.61 ± 0.21
**16**	4.21 ± 0.01	15.75 ± 0.15	165.51 ± 1.50	197.75 ± 3.75	4.55 ± 0.05	49.80 ± 0.65	26.82 ± 0.55	24.65 ± 0.12
**17**	1.25 ± 0.02	5.41 ± 0.71	23.15 ± 1.15	31.35 ± 7.50	1.45 ± 0.01	18.55 ± 0.90	9.20 ± 0.05	11.95 ± 0.05
**18**	4.75 ± 0.15	21.70 ± 0.10	285.02 ± 2.05	360.31 ± 1.12	4.52 ± 0.11	77.85 ± 0.45	46.95 ± 1.22	59.25 ± 1.13
**Total**	**92.50 ± 1.66**	**300.80 ± 0.52**	**1434.80 ± 2.82**	**2356.80 ± 2.18**	**64.35 ± 1.14**	**1668.90 ± 1.18**	**2197.25 ± 3.34**	**2404.30 ± 2.38**

mean values ± SD (*n* = 3), tr – traces under LOD.

**Table 6 molecules-24-04074-t006:** The content of saponins in red and white wines enriched with *Primula veris* L. (mg/L).

*Primula veris* L. (%)	Regional			Commercial		
	Red Wine					
	2.5	5	10	2.5	5	10
**Saponins**						
**1**	28.2 ± 0.29	43.35 ± 0.15	88.75 ± 0.23	25.30 ± 0.05	31.40 ± 0.25	64,70 ± 0.19
**2**	111.4 ± 0.42	161.85 ± 0.50	318.90 ± 1.93	75.65 ± 0.33	134.10 ± 0.69	222.90 ± 0.26
**3**	9.35 ± 0.07	16.55 ± 0.13	30.65 ± 0.29	2.90 ± 0.01	5.70 ± 0.05	10.00 ± 0.11
**Total**	148.95 ± 0.78	221.70 ± 0.78	438.35 ± 2.45	103.90 ± 0.27	171.25 ± 1.00	297.60 ± 0.56
	**White Wine**					
**1**	23.00 ± 0.09	34.70 ± 0.39	51.15 ± 0.31	24.80 ± 0.06	39.80 ± 0.34	52.20 ± 0.27
**2**	102.45 ± 0.05	124.00 ± 0.50	226.05 ± 1.71	87.15 ± 0.12	118.70 ± 0.96	199.35 ± 0.41
**3**	6.35 ± 0.07	8.70 ± 0.01	22.45 ± 0.06	8.25 ± 0.04	17.65 ± 0.02	19.25 ± 0.16
**Total**	131.75 ± 0.21	167.40 ± 0.90	299.65 ± 2.07	120.20 ± 0.10	173.20 ± 1.32	270.80 ± 0.01

mean values ± SD (n = 3).

**Table 7 molecules-24-04074-t007:** The content of flavonols and saponins in ethanol solvents enriched with *Primula veris* L. (mg/L).

*Primula veris* L. (%)	5		
Etanol (%)	40	70	96
**Flavonols**			
**1**	tr	tr	tr
**2**	tr	tr	tr
**3**	tr	tr	tr
**4**	tr	tr	tr
**5**	1.35 ± 0.02	tr	tr
**6**	7.30 ± 0.01	1.05 ± 0.02	0.20 ± 0.02
**7**	tr	tr	tr
**8**	2.50 ± 0.27	0.55 ± 0.01	0.30 ± 0.00
**9**	0.80 ± 0.00	tr	tr
**10**	3.80 ± 0.00	0.45 ± 0.00	0.20 ± 0.00
**11**	5.25 ± 0.00	0.80 ± 0.00	0.30 ± 0.00
**12**	24.4 ± 0.15	3.60 ± 0.01	1.80 ± 0.02
**13**	2.05 ± 0.00	0.25 ± 0.00	0.40 ± 0.01
**14**	11.20 ± 1.63	7.55 ± 0.04	0.40 ± 0.01
**15**	tr	tr	tr
**16**	27.95 ± 0.06	4.25 ± 0.03	2.70 ± 0.01
**17**	6.30 ± 0.01	1.20 ± 0.00	0.55 ± 0.00
**18**	52.75 ± 1.64	11.25 ± 0.05	3.90 ± 0.00
**Total**	145.70 ± 3.09 ^C^	27.65 ± 0.01 ^B^	11.00 ± 0.01 ^A^
**Saponins**			
**1**	34.35 ± 0.36 ^A^	37.65 ± 0.08 ^B^	93.15 ± 0.40 ^C^
**2**	117.90 ± 0.44 ^A^	167.65 ± 0.35 ^B^	317.65 ± 0.77 ^C^
**3**	16.25 ± 0.13 ^A^	23.25 ± 0.12 ^B^	33.00 ± 0.07 ^C^
**Total**	168.50 ± 0.92 ^A^	228.55 ± 0.31 ^B^	443.80 ± 0.44 ^C^

mean values ± SD (n = 3), tr – traces under LOD; Statistically significant differences between means (^A–C^ for *p* ≤ 0.01), marked by different letters in the rows.

**Table 8 molecules-24-04074-t008:** Influence of type of wine and the level of *Primula veris* L. addition on the profile of polyphenolic compounds and antioxidant properties, ethanol, and sugars content in enriched wines. ABTS: 2,2-azino-bis(3-ethylbenzothiazoline-6-sulfonic acid), FRAP: ferric reducing antioxidant power method, TPC: total phenolic compounds.

		WINES				*Primula veris* L. (%)			
		RED		WHITE		0	2.5	5	10
		Regional	Commercial	Regional	Commercial				
**mg/L**	**Total Anthocyanins**	1223.20 ^B^	308.20 ^A^	-----	-----	290.49 ^A^	300.14 ^B^	423.14 ^C^	517.00 ^D^
	**Total Flavonols**	900.18 ^B^	737.75 ^A^	1037.40 ^C^	1591.70 ^D^	79.73 ^A^	737.30 ^B^	1435.90 ^C^	2013.40 ^D^
	**Total Saponins**	202.25 ^C^	143.19 ^A^	149.70 ^B^	141.05 ^A^	-----	126.20 ^A^	183.38 ^B^	326.60 ^C^
**FRAP**		14.50 ^D^	10.47 ^C^	2.15 ^A^	2.44 ^B^	5.07 ^A^	6.76 ^B^	8.25 ^C^	9.50 ^D^
**DPPH**		11.76 ^D^	8.82 ^C^	1.67 ^A^	1.82 ^B^	4.58 ^A^	5.32 ^B^	6.74 ^C^	7.46 ^D^
**ABTS**		22.58 ^D^	16.67 ^C^	3.08 ^A^	3.67 ^B^	9.04 ^A^	10.27 ^B^	12.76 ^C^	1393 ^D^
**TPC**		1577.80 ^D^	932.35 ^C^	431.51 ^B^	359.64 ^A^	482.97 ^A^	726.62 ^B^	898.38 ^C^	1193.30 ^D^
**Ethanol (%*v*/*v*)**		10.14 ^D^	9.91 ^B^	10.56 ^C^	9.38 ^A^	11.33 ^D^	10.34 ^C^	9.31 ^B^	8.97 ^A^
**g/L**	**Glucose**	0.11 ^A^	0.27 ^C^	0.33 ^D^	0.23 ^B^	0.08 ^A^	0.33 ^D^	0.23 ^B^	0.29 ^C^
	**Fructose**	0.33 ^B^	0.21 ^A^	0.43 ^C^	0.34 ^B^	0.18 ^A^	0.54 ^D^	0.26 ^B^	0.32 ^C^
**Total Sugars**		0.43 ^A,a^	0.48 ^b^	0.75 ^D^	0.56 ^C^	0.26 ^A^	0.87 ^D^	0.48 ^B^	0.61 ^C^

Statistically significant differences between means (^A–D^ for *p* ≤ 0.01; ^a–b^ for *p* ≤ 0.05), marked by different letter in the rows.

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
