# Peer review of "Assessment of the Bioactive Compounds in White and Red Wines Enriched with a Primula veris L."

_molecules, 2019, doi:10.3390/molecules24224074_

Round 1
Reviewer 1 Report
The article needs more revisions to be published.The introduction should be rewritten focusing on the phenolic compounds and the mechanisms that cause the increase of these compounds by the addition of the herb studied.
Although the results are conclusive of a large increase in phenolic compounds, it remains to be incorporated as it affects the aroma of the wines. This is very important, since an alteration of the aroma can cause a rejection by the consumer.
It is convenient to provide a sensory analysis of the wines and the levels of the main aromas.
Author Response
Dear Reviewer,
Thank you very much for Your constructive comments and important remarks.
We have tried to make changes and corrections in the text according to Your recommendations.
Point 1: The article needs more revisions to be published.The introduction should be rewritten focusing on the phenolic compounds and the mechanisms that cause the increase of these compounds by the addition of the herb studied.
Response 1:
The introduction has been improved in the manuscript focusing on the content of phenolic compounds by the herbs addition.
Point 2: Although the results are conclusive of a large increase in phenolic compounds, it remains to be incorporated as it affects the aroma of the wines. This is very important, since an alteration of the aroma can cause a rejection by the consumer
Response 2:
The addition of herbs to wine definitely affects the aroma of wine. This is a very important and valuable remark and thank you very much for it. In our research, the wine was analyzed in terms of health potential only, without the quality control of final products. However, it is planned to expand the analysis in the next stage of the implementation of our research Project and then aroma tests and sensory analysis of wine samples with the addition of Primula veris L. will be carried out. In addition, experiments are also planned on enriching wines with purified extracts of primrose and other morphological parts of this plant as well as flowers of Primula veris L. subjected to lyophilization. At this stage, we are not able to perform additional tests related to organoleptic assessment and volatile compounds profile on samples of regional and commercial wines enriched with Primula veris L. flower because these samples were used during ongoing analyzes. In addition, the flowering time for Primula veris L. falls in the early spring (April / May), therefore further studies are planned for the next Spring. We hope that further experiments will also give interesting results.
Point 3: It is convenient to provide a sensory analysis of the wines and the levels of the main aromas
Response 3:
We completely agree with the above remark, however, due to the flowering time of Primula veris L. in Poland, we are not able to carry out additional expanded research on the sensory quality of fortified wines of Primula veris L. They have been planned for next Spring.

Reviewer 2 Report
This paper deals with the assessment of the bioactive compounds in white and red wines enriched with a Primula veris L. There are some suggestions that the authors should consider to improve how the manuscript is presented.
Abstract line 17-19: “For white wines, a 5-fold increase in flavonol content was found in regional and wine samples, and an almost 25-fold increase in commercial wine samples at the lowest Primula veris L. flower addition of 2.5%. “ Please clarify better the name of the wine samples in the abstract, for example, the authors referred to regional and wine samples and commercial wine samples. In material and methods, it is clear but in the abstract, it is not understood
Line 98-99: “Based on MS chromatography analysis, profiles of compounds belonging to anthocyanins (24), flavonols (18) and saponins (3) have been identified in the white and red wine samples. “ Please separate the compounds from white from the compounds identified in red wine.
Table 3.: “The content of anthocyanins in red wines enriched with Primula veris L. (mg/L). “ How the authors explained that in regional wine the total concentration of anthocyanins increased with increasing the enriched with Primula veris L. and for commercial wine the opposite was observed?
Table 7. “The content of flavonols and saponins in ethanol solvents enriched with Primula veris L. (mg/L)” How the authors explained that increasing the ethanol concentration the content of flavonols extracted from Primula veris L. decreased?
Author Response
Dear Reviewer,
Thank you very much for Your constructive comments and important remarks. We have tried to make changes and corrections in the text according to Your recommendations.
Point 1: Abstract line 17-19: “For white wines, a 5-fold increase in flavonol content was found in regional and wine samples, and an almost 25-fold increase in commercial wine samples at the lowest Primula veris L. flower addition of 2.5%. “ Please clarify better the name of the wine samples in the abstract, for example, the authors referred to regional and wine samples and commercial wine samples. In material and methods, it is clear but in the abstract, it is not understood
Response 1:
The names of wine sampales in the abstract has been corrected in the manuscript.
Point 2: Line 98-99: “Based on MS chromatography analysis, profiles of compounds belonging to anthocyanins (24), flavonols (18) and saponins (3) have been identified in the white and red wine samples. “ Please separate the compounds from white from the compounds identified in red wine.
Response 2:
The compounds identified in red and white wine has been separeted in the description of results in manuscript.
Point 3: Table 3.: “The content of anthocyanins in red wines enriched with Primula veris L. (mg/L). “ How the authors explained that in regional wine the total concentration of anthocyanins increased with increasing the enriched with Primula veris L. and for commercial wine the opposite was observed?
Response 3:
The analyzed wines, both regional and commercial, were produced from other grape varieties. The regional wine has been made from hybrid varieties which are characterized by a richer profile of anthocyanin compounds in which diglycoside forms occur. Commercial wine has been produced from noble varieties, which are in turn characterized by the presence of monoglycoside forms. During fermentation and, especially in the first one or two years of maturation, the monomeric anthocyanins in wines undergo a wide variety of reactions and associations and various anthocyanin-derived new pigments are formed, which are extremely crucial for the color stability. Consequently, although the concentration of monomeric anthocyanins in red wines declines constantly, red wines can still maintain an essentially red color. The reactions and associations involve complex mechanisms, including relatively short-term ones, such as self-association and copigmentation, and the relatively long-term ones, such as the formation of polymeric anthocyanins with flavan-3-ols and proanthocyanidins, as well as the formation of new pigments, such as pyranoanthocyanins and their further polymerized products.
To the best of our knowledge the mechanisms of maturing and stabilizing hybrid wines are not fully understood. The addition of a primrose herb component can modify these mechanisms. In our opinion, the addition of the aforementioned herb can modify the pH of the wine and the level of acidity which could transform the occurring combinations into native forms.
It needs more investigation and in our opinion this is a good contribution to further analysis.
Point 4: Table 7. “The content of flavonols and saponins in ethanol solvents enriched with Primula veris L. (mg/L)” How the authors explained that increasing the ethanol concentration the content of flavonols extracted from Primula veris L. decreased?
Response 4:
Decreasing flavone concentration is probably associated with polarity and thus with the solubility of the analyzed compounds. Unlike aglycons, glycoside forms dissolve better in aqueous alcoholic solutions and more importantly, remaining in purely alcoholic solutions have a tendency to precipitate. The increasing concentration of ethanol probably affected the soluble and thus the concentration mainly of glycosidic forms.

Round 2
Reviewer 1 Report
For the investigation to be complete, they must conduct a study of the main aromas and a sensory analysis.